# Next Generation Sequencing Analysis of MODY-X Patients: A Case Report Series

**DOI:** 10.3390/jpm12101613

**Published:** 2022-09-30

**Authors:** Giulio Maltoni, Roberto Franceschi, Valeria Di Natale, Randa Al-Qaisi, Valentina Greco, Roberto Bertorelli, Veronica De Sanctis, Alessandro Quattrone, Vilma Mantovani, Vittoria Cauvin, Stefano Zucchini

**Affiliations:** 1Pediatric Unit, IRCCS AOU, S. Orsola-Malpighi, 40138 Bologna, Italy; 2Pediatric Unit, S. Chiara Hospital of Trento, 38122 Trento, Italy; 3Advanced Molecular Diagnostic Laboratory, Department CIBIO-DMA, University of Trento, 38123 Trento, Italy; 4Next Generation Sequencing Core Facility, LaBSSAH, Department CIBIO, University of Trento, 38123 Trento, Italy; 5Laboratory of Translational Genomics, Department CIBIO, University of Trento, 38123 Trento, Italy; 6Applied Biomedical Research Center, CRBA, S. Orsola-Malpighi, 40138 Bologna, Italy

**Keywords:** NGS, MODY-X, precision medicine

## Abstract

Background: Classic criteria for a maturity-onset diabetes of the young (MODY) diagnosis are often unable to identify all subjects, and traditional Sanger sequencing, using a candidate gene approach, leads to a high prevalence of missed genetic diagnosis, classified as MODY-X. Next generation sequencing (NGS) panels provide a highly sensitive method even for rare forms. Methods: We investigated 28 pediatric subjects suspected for MODY-X, utilizing a 15-gene NGS panel for monogenic diabetes (MD). Results: NGS detected variants of uncertain significance (VUS), likely pathogenic or pathogenic for rarer subtypes of MODY, in six patients. We found variants in the wolframin gene (*WFS1*), traditionally not considered in MD genetic screening panels, in three patients; *KCNJ11* gene mutation, typically responsible for neonatal diabetes and rarely causing isolated diabetes in adolescents; *INS* gene mutation; a variant in the *HNF1B* gene in a young male with diabetes on sulfonylurea treatment. Conclusion: In our cohort, the availability of an NGS panel for MD was determined for the correct identification of MD subtypes in six patients with MODY-X. Our study underlines how a precise diagnosis utilizing NGS may have an impact on the management of different forms of MODY and, thus, lead to a tailored treatment and enable genetic counselling of other family members.

## 1. Introduction

Monogenic diabetes (MD) is a rare form of diabetes, accounting for approximately 1 to 6% of pediatric diabetes patients [1,2]. Most cases of MD are represented by maturity-onset diabetes of the young (MODY), characterized by early hyperglycemia onset for defects in insulin secretion and negative beta-cell autoantibodies [3]. Identification of children with MODY is fundamental in order to ensure the most appropriate management and follow-up as well as to enable genetic counseling of other family members [4,5].

However, classic clinical criteria for MODY are often unable to identify all the cases; around 80–90% of MODY subjects are misdiagnosed [6,7]. Definitive diagnosis relies on genetic tests: the traditional Sanger sequencing, using a candidate gene approach, leads to a high prevalence (46.2–73.9%) of unclear genetic diagnosis classified as MODY-X [8,9,10,11,12], whereas targeted next generation sequencing (NGS) panels, enabling a simultaneous analysis of multiple genes, provides a highly sensitive method even for rare forms [13]. This is a collection of case reports that aims to show how a precise diagnosis via NGS could lead to a tailored treatment and an appropriate follow-up, according to a precision medicine approach.

## 2. Subjects and Methods

### 2.1. Subjects

We decided to screen patients with a failed diagnosis of MODY by Sanger sequencing (suspected MODY-X), utilizing NGS gene panels. Inclusion criteria included the following:(1)Patients from 0 to 18 years with confirmed persistent hyperglycemia (two or more occasions) and/or diabetes onset in the period from January 2000 to December 2016, recruited at the Pediatric Diabetic Clinic, S. Orsola Malpighi, Bologna and S. Chiara Hospital in Trento;(2)Absence of an autoimmune marker: glutamic acid decarboxylase antibodies (GAD), insulin antibodies (IAA), autoantibody to protein tyrosine phosphatase (IA-2), autoantibody to the cation efflux transporter zinc transporter (ZnT8);(3)One first-degree relative diagnosed with hyperglycemia or diabetes before age 25 years (non-binding criterion for inclusion in the study);(4)Diabetes treatment with diet and/or oral agents and/or reduced insulin need after at least 1 year from diabetes onset;(5)Negative results of genetic analysis of the most prevalent types of MODY in Italy (*GCK*-, *HNF1A*-, and *HNF4A*-MODY).

According to this criteria, 28 subjects with MODY-X were enrolled in this study and analyzed by NGS. The study was approved by the Hospital Ethics Committee of S. Orsola-Malpighi Hospital, Bologna and S. Chiara Hospital, Trento. The patients and/or legal guardians provided a written informed consent form before their enrollment.

### 2.2. Methods

A 15-gene NGS panel for MD (+ 25 bp in the close intronic regions), investigating for *HNF4A*, *GCK*, *HNF1A*, *PAX4*, *HNF1B*, *NEUROD1*, *APPL1*, *KLF11*, *CEL*, *BLK*, *PDX1*, *ABCC8*, *KCNJ11*, *INS*, and *WFS1*, was performed. All the frameshift, nonsense mutations and those on canonical splicing sites were considered pathogenic; for all the missense variants, the presence in two HGMDs (Human Gene Mutation Database) and LOVD3.0 (Leiden Open Variation Database), which collect pathogenic mutations and classify them according to literature data, were verified. The variants were classified according to the criteria established in 2015 by the American College of Medical Genetics and Genomics (ACMG) [14], in five categories: pathogenetic, probable pathogenetic, of uncertain significance, probable benign, and benign, according to the ClinVar reports.

To evaluate the frequency in the general population, the ExAC (Exome Aggregation Consortium) database was consulted: this reports MAF (minor allele frequency = frequency of the minor allele) data, calculated on a population of 60,706 unrelated cases. Lastly, all the missense variants were subjected to bioinformatic tools of pathogenic prediction (PolyPhen 2 and SIFT sequence).

## 3. Results

The family history, clinical, and biochemical features of the six patients who were identified as positive after NGS for a variant of uncertain significance (VUS), likely pathogenic or pathogenic, were reported. A summary is presented in Table 1 [15,16,17,18,19,20]. Six other patients showed benign or likely benign variants after NGS analysis, and sixteen showed a negative result.

**Case 1:** An 18-year-old boy was referred to our center for mild to moderate hyperglycemia (110–170 mg/dL) since adolescence, with HbA1c levels between 6 and 6.5% (42–47 mmol/mol). Autoantibodies for type 1 diabetes (T1D) were negative. Due to the family history of diabetes (mother and grandmother on treatment with oral hypoglycemic drugs), genetic tests for the most common forms of MODY were performed (*GCK*-, *HNF1A*-, and *HNF4A*- MODY) and found negative. He then underwent an NGS panel test, and a *WFS1* (wolframin) stop codon variant p.Arg42* (c.124C > T), classified as pathogenic, was found. The same mutation was identified in his mother. As his metabolic control was acceptable (HbA1c constantly <7%, 53 mmol/mol), no treatment was started.

**Case 2:** A 10-year-old boy was admitted for sustained hyperglycemia (blood glucose > 300 mg/dL, HbA1c > 9%, 75 mmol/mol) without DKA. Given the age and the consistent insulin requirement (0.8 IU/kg/day), he was initially diagnosed as T1D. All autoantibodies for beta cells were negative. Remarkably, he was under treatment in another center for congenital cataract as his mother had gradually developed mild to moderate hyperglycaemia.

In the absence of autoantibodies and his family history, he was also investigated for the main types of MODY (*GCK*-, *HNF1A*-, and *HNF4A*- MODY), but no mutation was found. Subsequently, he underwent an NGS panel test and a genetic variant in exon 8 of *WFS1* gene (c.1153G > A, p.Glu385Lys), classified as VUS, was identified. The PolyPhen score was 0.96 (damaging), and the SIFT was 0.077 (tolerated).

This variant was considered pathogenetic because computational prediction tools unanimously support a deleterious effect on the gene, even if the allele frequency is greater than expected for the disorder (ƒ = 0.0012). We decided to test his mother, as she presented hyperglycaemia, and the same missense mutation in *WFS1* gene was found.

**Case 3:** A 12-year-old girl was referred for diabetes with mild DKA and negative T1D specific antibodies. Her HbA1C at diagnosis was 16.8% and insulin treatment (0.5 IU/kg/day) was started. Basal C-peptide was 1.3 ng/mL and 3.8 ng/mL after glucagon stimulation, consistent with a good residual β-cell function. A predisposing HLA to T1D (DR3 and DR4) was found. After 2 years, rapid insulin was discontinued and therapy with basal insulin (0.3–0.4 U/kg/d) and metformin was started due to the low daily insulin requirement and recurrent hypoglycaemia. Her HbA1C ranged between 6.5% and 7.2% (48–55 mmol/mol). Her father, affected by diabetes, was treated with basal insulin and metformin. 

After 3 years from diabetes onset, the younger sister also presented high fasting blood glucose, which was detected early thanks to family screening tests. She showed positivity of antibodies specific for T1D (GAD Ab 1181 UI/mL, ICA positive) in association with autoimmune thyroiditis (TPO Ab 51.6 UI/mL, TG Ab 94 UI/mL), and she was diagnosed as having T1D. She also presented predisposing HLA to T1D like her sister. Her basal C-peptide at diagnosis was 0.6 ng/mL, and her daily insulin requirement during follow-up was 0.7 IU/kg/day.

Given the successful “hybrid” treatment (both insulin and metformin) and her family history for diabetes, the Sanger genetic test for the principal forms of MODY was performed, but no mutations were found. Subsequently, with the possibility to utilize the NGS panel, we decided to perform genetic analysis, and two allelic variants were detected: p.Leu432Val c.1249T > G in exon 8 on the wolframin gene and p.Ala71Thr c.211G > A in exon 4 on the BLK gene.

For the variant in the *WFS1* gene, PolyPhen was 0.076 (prediction: benign), and SIFT was 0.018 (prediction: damaging), whilst the missense variant in the *BLK* gene has currently been reclassified as benign/likely benign.

The sister affected by T1D and the father were also tested. The sister showed only the variant in the *WFS1* gene, whilst the father harbored both *WFS1* and *BLK* variants.

**Case 4:** An adolescent presented with very high blood glucose (>400 mg/dL) and HbA1c 8.9% (74 mmol/mol) at diagnosis with negative T1D specific antibodies. At the age of 16 years old, his father was diagnosed with DM, generically labelled as insulin-dependent, even if initially treated with oral hypoglycaemic agents and then shifted to insulin treatment. After an initial period on insulin treatment, for the rapid decreasing insulin requirement, the subject was put on sulfonylurea treatment (glimepiride) with a good metabolic response and HbA1C ranging from 6.5 to 7.5% (48–58 mmol/mol). Thus, the Sanger genetic test for the principal forms of MODY was performed, but no mutations were found.

Subsequently, the NGS panel for monogenic diabetes was proposed, and a missense mutation in the *KCNJ11* gene was found (c.685 G > A; p.Glu229Lys), classified as likely pathogenic. The same mutation in the *KCNJ11* gene was shared by his father.

**Case 5:** A 15-year-old boy came to our attention for suspected T1D. His blood glucose was 436 mg/dL, and his HbA1c was 9.1%, 76 mmol/mol. He presented polyuria and polydipsia as well as weight loss.

The boy was the youngest of six siblings (two males and four females): the older brother had been on insulin therapy since the age of 17, a sister had diabetes since the age of 18, and the other 3 sisters had presented gestational diabetes and were still on oral hypoglycemic therapy. The mother was affected by T2D. Due to the marked family history for diabetes, a genetic investigation in the suspicion of MD was initially performed, using Sanger sequencing for *HNF4A*-, *HNF1A*-, and *GCK*-MODY, but no mutations were found. Subsequently, because the NGS method made it possible to analyze rarer genes, a splicing mutation in the preproinsulin gene (*INS*) was found (c.187 + 2T > C) and classified as pathogenic [21].

**Case 6:** A 13-year-old boy presented with a blood glucose of 491 mg/dL, HbA1c 8.6%, and negative T1D-specific antibodies. He needed only basal insulin of 0.66 IU/kg/day. The father of the proband was diagnosed with T2DM when he was 25 years old, and his paternal grandmother was also affected by T2DM. The Sanger genetic test for the principal forms of MODY found no mutations. By NGS, the patient was identified with a change in the *HNF1B* (hepatocyte nuclear factor-1B) gene (c.704G > A, p.Arg235Gln). This variant has previously been reported as pathogenic in the literature [21]. An abdominal ultrasound did not identify renal cysts, and an abdominal MRI did not reveal pancreatic body or tail agenesis; he had normal kidney and liver function. Therefore, he did not show typical features of MODY-5.

The father was tested and proved to be positive for the same mutation; he had a normal abdominal US. Although our patient did not show evident pancreatic/extra-pancreatic manifestations, a phenotypic correlation with the *HNF1B* genetic finding was established, as concluded for other patients in the literature [22].

## 4. Discussion

In our study, the use of an NGS panel for monogenic diabetes provided new pathogenic diagnoses for rarer subtypes of MODY in a cohort of subjects previously labeled as MODY-X, on the suspicion of MD without a genetically confirmed diagnosis.

Cases 1, 2, and 3 carried variants in the *WFS1* gene. *WFS1* encodes a transmembrane protein, wolframin, primarily involved in membrane trafficking and endoplasmic reticulum homeostasis. Although homozygous or compound heterozygous mutations in the *WFS1* gene are the main causes of Wolfram syndrome, it is less known whether common variants in *WFS1* could confer a higher risk of type 2 diabetes [15]. More recently, some case reports or case series have been published, detailing non-syndromic and non-autoimmune insulin-dependent forms of diabetes due to the *WFS1* mutations [23]. Moreover, carriers of the *WFS1* pathogenic variants can present with features unique to the Wolfram syndrome, such as congenital cataract, non-syndromic sensorineural deafness, or psychotic disorder [24], and these characteristics could be helpful in leading to a clinical suspicion. Traditionally, *WFS1*-related diabetes was not considered in MD genetic screening panels as they were usually based only on MODY genes.

Case 3 showed that the investigation of multiple genes, made possible with NGS panels, allows more variants at a time to be found, but, as for the *BLK* gene in this patient [25], it is unlikely it played a role in causing the disease. An additional problem with the use of NGS is linked to the classification of variants into benign, likely benign, VUS, likely pathogenic, and pathogenic as well as how to report them. Notwithstanding the ACMG clinical classification guidelines [12], some authors reported that because of the paucity of cases, it might be difficult to identify the pathophysiologic mechanism of disease in new genic variants [26].

Case 4 carried a KCNJ11 gene mutation, typically responsible for neonatal diabetes, but heterozygous mutations in this gene rarely cause isolated diabetes in children and adolescents, and some authors recommend that the molecular diagnosis of MODY should include the *KCNJ11* gene [26]. Pathogenic variants in *ABCC8* and *KCNJ11* genes, which encode the subunits of the β-cell ATP-sensitive potassium channel, may positively respond to oral agents, such as sulfonylureas, as reported in our patient [27].

Case 5 revealed a *INS* gene mutation. *INS* encodes preproinsulin, a single chain precursor molecule, post-translationally converted to insulin in the pancreatic b-cells [28]. Mutations in the *INS* gene can be associated with a broad spectrum of clinical presentations, varying from intrauterine insulin deficiency and permanent neonatal diabetes to mild young- and adult-onset diabetes [29]. Members of the same family sharing the same mutation in the *INS* gene may present different clinical phenotypes, and environmental or epigenetic factors may contribute to the pathogenesis of INS-MODY [30].

Case 6 was diagnosed as MODY5. This condition is caused by mutations in the *HNF1B* gene encoding the HNF1B transcription factor. Patients with HNF1B-MODY often have additional extra-pancreatic features, such as renal and/or hepatic cysts, elevated liver enzymes, genitourinary malformations, mental retardation, or eye defects [31]. A wide clinical spectrum has been reported in MODY5 patients: the first recorded manifestation of the disease in 70% of patients is diabetes, with or without associated renal disease [20]. Extra-pancreatic symptoms have been reported in 40% of patients with HNF1B-MODY in the DPV database [32]. The possibility of an NGS panel for MD allowed us to test the subject for HNF1B-MODY, despite the rather weak clinical suspicion.

## 5. Conclusions

The development of novel tests based on NGS can allow a molecular diagnosis at affordable costs and timing, performing the simultaneous analysis of multiple genes. In our cohort, the use of a gene panel was a pivotal factor for the correct identification of monogenic diabetes subtypes in many cases. The genetic diagnoses of these patients could have been missed due to the absence of typical clinical findings during the first evaluation and the negative results of the traditional gene-by-gene approach for the most common causes of MD.

This study underlines the fact that the application of precision medicine in the diagnosis and treatment of monogenic diabetes is a standard of care [33]. Precision diagnostics with NGS and tailored treatments have an impact on the management of different forms of MODY, also in other affected family members. Greater availability and lower costs over time will overcome the problem of the accuracy of probabilistic algorithms or calculators that consider family history, clinical, and biochemical features as a means to identify patients who would be candidates for NGS. In MD, NGS is preferable to Sanger sequencing, which can be ineffective, time-consuming, and result in a misdiagnosis in subjects with overlapping symptoms in the absence of a typical phenotype.

## Figures and Tables

**Table 1 jpm-12-01613-t001:** Summary of clinical, biochemical, and genetic data of the six patients who turned out as positive at NGS for a variant of uncertain significance (VUS), likely pathogenic or pathogenic. All the variants were inherited in heterozygosity.

Patient	Gender	Age at the Hyperglycemia	BG (mg/dL) HbA1c at First Observation	Phenotype	Therapy	Family History	Gene	Nucleotide Change	Amino Acid	Segregation	Allele Frequency (gnomAD)	PolyPhen-2 SIFT	ClinVar Classification	Literature Reports
1	M	18	110–170 6–6.5%	-	None	Mother and grandmother on OHA	*WFS1*	c.124C > T	p.Arg42Ter	Mother	0.00009	Pathogenic	Pathogenic/Likely pathogenic	[15]
2	M	10	300 >9%	Congenital cataract	Insulin 0.8 IU/kg/day	Mother with mild/moderate hyperglycemia	*WFS1*	c.1153G > A	p.Glu385Lys	Mother	0.00057	VUS	VUS/Likely benign	[16]
3	F	12	>400 16.8%	-	Metformin	Father with DM on insulin and metformin; sister on insulin	*WFS1* *BLK*	c.1249T > G c.211G > A	p.Arg42Ter p.Ala71Thr	Father, sister Father	0.01485	VUS benign	VUS Benign/Likely benign	[17]
4	M		400 8.9%	-	Sulfonylurea	Father with DM on insulin	*KCNJ11*	c.685G > A	p.Glu229Lys	Father	-	Likely pathogenic	Pathogenic/Likely risk allele	[18]
5	M	15	436 9.1%	-	Insulin	Brother and sister with diabetes; three sisters on OHA; mother with T2D	*INS*	c.187 + 2T > C	Intron variant	Mother, 2 brothers, and 4 sisters	-	Pathogenic	Pathogenic	[19]
6	M	13	491 8.6%	Normal abdomen US	Insulin 0.66 IU/kg/day	Father and grandmother with T2D	*HNF1B*	c.704G > A	p.Arg235Gln	Father and grandmother	-	Pathogenic	Likely pathogenic/VUS	[20]

BG: blood glucose, HbA1c: glycated hemoglobin, T2D: type 2 diabetes, OHA: oral hypoglycemic agent.

## Data Availability

The data that support the findings of this study are not publicly avail- able because they contain information that could compromise the privacy of research participants but are available from the corresponding author upon reasonable request.

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
