# Peer review of "Next Generation Sequencing Analysis of MODY-X Patients: A Case Report Series"

_jpm, 2022, doi:10.3390/jpm12101613_

Round 1

Reviewer 1 Report

Maltoni et al present a case series of MODY-X cases with molecular genetic bases identified by panel NGS testing.  Increasing the recognition of MODY is extremely valuable, although their discussion of the value of panel NGS testing is may be of limited regional interest since panel NGS for MODY is standard in many parts of the world.  The manuscript is also somewhat limited by non-standard use of English. 

Abstract: In the first sentence of the abstract, it is unclear whether the authors are criticizing MODY classical criteria, or the lack of identification of the causative mutations by targeted Sanger testing. 

Line 36-37: Likewise, are the problem with the criteria, lack of recognition of MODY, or inability to detect a molecular basis of disease using targeted testing.  The authors need to be clear in their arguments.

Lines 66-78: Since the authors are using the ACMG nomenclature of pathogenic, likely pathogenic, VUS, likely benign, and benign, they need to cite the paper outlining those categories and to apply them to their categorization of variants (Richards, Aziz, Bale et al).  In addition, presenting Polyphen & SIFT data (especially to VUS) would be useful.

Line 81 and others: The phrase “turned out as” sounds unprofessional.  Perhaps versions of the phrase “testing identified…” could be used as a replacement.

Case 2: The ACMG criteria for this WFS1 VUS need to be outlined and/or Polyphen & SIFT data discussed.  Segregation in the family is good, but with a single generation it is not particularly convincing. Use of the term ‘mutation’ is misleading as it suggests pathogenicity—genetic variant is the accurate ter

Case 4: The idea of an interaction and/or bigenic inheritance of risk is intriguing, but not convincing.  Did the proband have the benign BLK1 variant?  What is the evidence that a benign variant would have any functional effect on the protein?  It is also complicated in this case by the sister’s more typical (?) T1D diagnosis and their permissive HLA type.  I would leave this case out or, remove the BLK1 discussion and present Polyphen & SIFT predictions for the WFS1 VUS.

Lines 157-159: This sentence is run-on.  Perhaps: “This variant has previously been reported as pathogenic in the literature.  No renal cysts were identified by abdominal ultrasound, abdominal MRI did not reveal pancreatic body or tail agenesis, and he had normal kidney and liver function.  Therefore, he did not show typical features of MODY 5.”

Line 170: Use the traditional spellings for “homozygous” and “heterozygous”

Line 174-176: Sentence needs work. Perhaps: “Moreover, carries of WFS1 pathogenic variants can present with features unique from Wolfram syndrome, such as…”

Line 180: “Case 3 carried a KCNJ11…”

Lines 194-196: The problem of how to classify and how to report genetic variants is well-addressed. Cite the ACMG clinical classification guidelines.

Lines 225-227: I would argue against throwing out clinical calculators altogether (lines 225-227) since NGS panels like the authors’ fail to identify >75% of MODY-X cases.

Overall organization: Please group the WFS1-implicated cases and start with the more traditional HNF1B and INS-implicated cases.  Use a table to present a summary of clinical data.  Avoid extraneous arguments unless they are fully developed—e.g. criticism of MODY criteria (line 13 & 36) or problems with variant classification (line 195).

A native English editor also needs to read and edit this manuscript.

Reviewer 2 Report

Aug 2022

Review of Maltoni et al.

J. Pers. Med- 1888866

In this paper the authors’ address the problem that many of the patients with Maturity‐Onset Diabetes of the Young (MODY) are not diagnosed properly, using the traditional methods. This leads to insufficient treatment for them and potential family members. The solution suggested is using Next-Generation Sequencing (NGS) panels which is a highly sensitive method even for rare forms of MODY. 

In this case report series, the distinct mutation in a specific gene was detected in six patients, and which was also present in at least one more family member. This enabled personalized tailored treatments in addition to genetic counselling to other family members, hence suggests applying this method on a routine basis in unidentified patients.

Below you will find my comments.

1. The manuscript requires careful review and editing for English and for style.

2. When describing the cases be consistent and include for all how they fit the inclusion criteria, what treatment they received etc.

3. Line 23 – "In our cohort NGS was decisive for the correct identification of MD subtypes in 6 patients with MODY X". 

4. Line 56 – "4) the proband and at least one first-degree relative diagnosed with hyperglycemia or diabetes before age 25 years". 

It is not clear that the authors followed their own inclusion criteria. For instance, in case number 1 they mention "mother and grandmother on treatment with oral hypoglycemic drugs" – were they actually diagnosed under the age of 25 as required in criteria 4.

5. Line 67 – "A 15-gene NGS panel for MD". 

On what basis were the 15 genes in the NGS were chosen?

6. Line 73-75 – "To evaluate the frequency in the general population, the ExAC (Exome Aggregation Consortium) database was consulted, which reports MAF (Minor Frequency Allele = frequency of the minor allele) data, calculated on a population of 60.706 cases not related". 

I failed to find where the results of this analysis are included in the report. 

Should "variant of uncertain significance (VUS)" (line 81), be considered decisive?

7. Line 87- 89 – "As a family history of diabetes (mother and grandmother on treatment with oral hypoglycemic drugs), genetic tests for the most common forms of MODY were performed (GCK-, HNF1A- and HNF4A MODY) and found negative". 

This sentence is not clear.

8. Line 101-103 – "Subsequently, he underwent a NGS panel test and a mutation 101

in a WFS1 variant (c.1153G>A, p.Glu385Lys), classified as VUS, was identified and so we decided to test the mother herself and the same mutation in WFS1 gene was found".

Consider rephrasing this sentence.

9. Line 106 – "His father had DM what type?" Should this be T2DM? or perhaps a typo and should be MD?

10. Line 152 – "The father of the proband as well as paternal grandmother presented with T2DM". At what age?

11. Lines 206-213 – "Case 6 …" 

What about relating the information you bring in the discussion to your patient?

12. Line 217 – "was a decisive factor for the correct identification of monogenic diabetes subtypes in most cases". Regarding the word decisive see previous comment. Regarding most cases to my understanding, you tested 27 cases and found factors for 6. Is 6/27 considered most?

13. Screen the manuscript for format and typo such as "Paediatric" on line 17 or space between lines in lines 62-65.

Reviewer 3 Report

The manuscript by Maltoni et al reported their experience on NGS MODY panels in diagnosis in 28 patients, with six of them of uncertain or positive test results. The selection criteria of patients are clearly established and six case reports are described.

Major

The authors have stated the classification of each diagnosis/best candidate variant in the manuscript. However, it is well known that variant classification is subjective to manual experience and judgement. I would suggest the authors providing more variant information to the readers, for example, the population occurrence, literature reports, ClinVar classification, prediction score, etc. Segregation analysis was already described in the text, but putting together with other info, it could provide a better picture. A summary in table format would be clear and concise, but brief description could be good too.

Minor

Line 17: “suspect” should be “suspect for”

Round 2

Reviewer 1 Report

Line 81: while the authors indicate that they classify variants per the 2015 ACMG criteria, they actually report the ClinVar classification performed by others.  This is reasonable, but their methods should reflect this.

Line 128: the word ‘with’ is missing.

Lines 142-143, 210-212: the benign/likely benign variant in BLK doesn’t track with disease in the family, is not thought to affect function (at least not in a penetrant manner) and is unlikely to play a role here and confuses the issues.  It should be removed. 

Line 215-217: the meaning of this sentence remains unclear.  Are they referring to identifying new disease associations or to assigning low penetrance/polygenic effects to common genetic variants.

There are spacing errors introduced by tracking changes.  Careful editing should identify these. 

Author Response

We would like to thank again the reviewer for his/her comments that contributed to improve the quality of our study. Herewith follow our responses:

REVIEWER 1:

Line 81: while the authors indicate that they classify variants per the 2015 ACMG criteria, they actually report the ClinVar classification performed by others.  This is reasonable, but their methods should reflect this.

A: the text has been modified as suggested.

Line 128: the word ‘with’ is missing.
A: the sentence has been modified as suggested.

Lines 142-143, 210-212: the benign/likely benign variant in BLK doesn’t track with disease in the family, is not thought to affect function (at least not in a penetrant manner) and is unlikely to play a role here and confuses the issues.  It should be removed. 

A: the text has been modified as suggested.

Line 215-217: the meaning of this sentence remains unclear.  Are they referring to identifying new disease associations or to assigning low penetrance/polygenic effects to common genetic variants.

A: the text has been modified as suggested.

There are spacing errors introduced by tracking changes.  Careful editing should identify these. 

A: the text has been modified as suggested.

Reviewer 2 Report

The manuscript still requires careful review and editing for English and for style.

Author Response

We would like to thank again the reviewer for his/her comments that contributed to improve the quality of our study. Herewith follow our responses:

The manuscript still requires careful review and editing for English and for style.

The text has been revised and edited for English language.